# The Gut Microbiome in the First One Thousand Days of Neurodevelopment: A Systematic Review from the Microbiome Perspective

**DOI:** 10.3390/microorganisms12030424

**Published:** 2024-02-20

**Authors:** Nathalia F. Naspolini, Paulo A. Schüroff, Maria J. Figueiredo, Gabriela E. Sbardellotto, Frederico R. Ferreira, Daniel Fatori, Guilherme V. Polanczyk, Alline C. Campos, Carla R. Taddei

**Affiliations:** 1School of Arts, Sciences and Humanity, University of São Paulo, São Paulo 03828-000, Brazil; nfnaspolini@usp.br (N.F.N.); p.schuroff@usp.br (P.A.S.); 2Department of Pharmacology, Ribeirão Preto Medical School, University of São Paulo, São Paulo 14049-900, Brazil; zzfigueiredo77@gmail.com; 3Department of Clinical and Toxicological Analyses, School of Pharmaceutical Sciences, University of São Paulo, São Paulo 05508-000, Brazil; 4Oswaldo Cruz Foundation, Oswaldo Cruz Institute, Av. Brasil, 4365, Manguinhos, Rio de Janeiro 21040-360, Brazil; frederico.ferreira@ioc.fiocruz.br; 5Department of Psychiatry, Medical School FMUSP, University of São Paulo, São Paulo 05403-903, Brazil; 6Division of Clinical Laboratory, University Hospital, University of São Paulo, São Paulo 05508-000, Brazil

**Keywords:** microbiome–gut–brain axis, early life microbiome, neurodevelopmental outcomes, microbial mechanism, *Bacteroides*, *Bifidobacterium*

## Abstract

Evidence shows that the gut microbiome in early life is an essential modulator of physiological processes related to healthy brain development, as well as mental and neurodegenerative disorders. Here, we conduct a systematic review of gut microbiome assessments on infants (both healthy and with conditions that affect brain development) during the first thousand days of life, associated with neurodevelopmental outcomes, with the aim of investigating key microbiome players and mechanisms through which the gut microbiome affects the brain. *Bacteroides* and *Bifidobacterium* were associated with non-social fear behavior, duration of orientation, cognitive and motricity development, and neurotypical brain development. Lachnospiraceae, *Streptococcus*, and *Faecalibacterium* showed variable levels of influence on behavior and brain development. Few studies described mechanistic insights related to NAD salvage, aspartate and asparagine biosynthesis, methanogenesis, pathways involved in bile acid transformation, short-chain fatty acids production, and microbial virulence genes. Further studies associating species to gene pathways and robustness in data analysis and integration are required to elucidate the functional mechanisms underlying the role of microbiome–gut–brain axis in early brain development.

## 1. Introduction

Human brain development starts during gestation with the proliferation of neural progenitor cells, followed by their fate, commitment, migration, differentiation, and the integration of the newly generated cells into the circuitries located in specific brain layers and subregions [1]. This process continues throughout our lifetime [2], especially represented by mechanisms related to the formation of new or more efficient synapsis, dendritic remodeling, and pruning, mechanism of myelination, synaptic pruning, myelination, and the formation of new neurons in postnatal neurogenic niches [3]. 

Signs coming from outside of the central nervous system (CNS) can orchestrate several brain functions, including metabolic, immune-mediated responses, and neuroplasticity. The brain–gut–microbiome axis seems to be one of the major players in the communication between the developing brain and the periphery, shaping and mediating brain plasticity and function. Previous studies have suggested that during brain development [4], the gut microbiome is essential for the formation and integrity of the blood–brain barrier (BBB), as well as for the processes of neurogenesis, microglia maturation, myelination, neurotrophins expression, and the fine tune production of neurotransmitters, and their respective receptors [5].

The gut colonization by microorganisms (the gut microbiota) begins in early life. In a certain way, it follows the rhythm of the brain’s postnatal development. Immediately after birth, the microbiota’s diversity gradually increases until a stable ecosystem is established, which resembles the adult type, adapting to the environments and habits of the host [6]. This crucial period, when homeostasis is being established, cross-communicate with the development of physiological processes, such as the immune system [7] and the brain [8] and can define how this interrelationship will progress [9].

The brain–gut–microbiome crosstalk includes different arms working together in a bidirectional direct or indirect pathway, involving the vagus nerve, the immune system, the hypothalamic–pituitary–adrenal axis, gut neurotransmitters, and neuromodulators. Bacterial products or metabolites including short-chain fatty acids (SCFAs) [10] can reach the brain and regulate brain functions such as cognition and stress-related responses [11,12].

Several reviews have reported on gut microbiome and neurobehavioral outcomes such as autism [13], attention deficit/hyperactivity disorder [14] depression/anxiety [15], and neurodevelopmental outcomes [16,17]. However, despite the evolving evidence linking the gut microbiome with brain and behavioral outcomes, the existing reviews lack exploring microbial mechanisms and their own understanding of the field.

To address this gap, we carried out a systematic review to examine the current evidence linking the infant gut microbiome with neurodevelopmental outcomes. We focused on describing the key microbial taxa affecting neurodevelopment and we further discussed the possible microbial mechanisms underlying these associations.

## 2. Methods

### 2.1. Search Process

We conducted a systematic review according to the parameters established by Page et al. (2020) [18]. The studies included were retrieved from searches made in the scientific bases LILACS, MEDLINE, MEDCARIB, PAHO-IRIS, and WHOLIS, or based on the close examination of relevant articles found in the reference lists of the studies included in the present systematic review. The scientific database was consulted up to November 2023. No restrictions were applied regarding the date of publication of the study. The following matches of keywords were used: (gastrointestinal microbio* or gut microbio*) and (neurodevelopment or neurobehavior). As a first stage of the analysis, the articles retrieved from the initial search were selected by their titles and abstracts. They were then screened according to the inclusion and exclusion criteria (Figure 1—modified from PRISMA flow diagram). Exclusion criteria included non-epidemiological or non-RCT (randomized controlled trials) studies; outcome other than cognitive development, imaging, and behavior; microbiome assessment > 3 years of age; non-sequencing/pyrosequencing methodologies. Inclusion criteria included microbiome evaluation; validated scales for brain outcomes evaluation; microbiome evaluation up to 3 years of life, including healthy and unhealthy infants. All studies included in this systematic review determined gut bacterial composition using next-generation sequencing technologies. In addition, exposure was considered the microbiome evaluation during the first three years of life, and the outcome was defined as cognitive development, imaging, and behavior assessment. From a total of 1424 papers, 23 were selected for final inclusion in our systematic review. The selected articles were independently revised (as full manuscripts) by two of the authors, N.F.N. and P.A.S. The PRISMA checklist is provided in the Appendix A and the protocol was registered in Prospero (CRD42023458504).

### 2.2. Detailed Description of the Data Collection, Extraction, and Quality Control Method

Data from the studies and their details were extracted according to authorship (publication date, country, and study period), study type, study population (subjects, sample size), microbiome (sequencing method, targeted region, timepoints), clinical outcomes (definition, data collection, timepoints) and main findings (studies overview; key taxa, alpha and beta diversity related to neurodevelopmental). Studies were categorized and organized in tables (Appendix A). The main findings were represented graphically, and a schematic figure shows possible microbial function underlying the main findings. We presented associations adjusted for potential confounders and correlation analysis. In selected studies in which adjusted analysis was not performed, crude associations were included in the systematic review.

## 3. Results

### 3.1. Studies Overview

We found 23 peer-reviewed studies reporting early life microbiome-related outcomes and neurodevelopment outcomes that met the eligibility criteria of this systematic review. Studies were carried out in children aged 0–3 years (microbiome assessment), followed by assessment of neurodevelopmental outcomes at the age of 10. Study types included prospective cohort (*n* = 15), cross-sectional (*n* = 4), case–control (*n* = 2), and RCT (*n* = 2). Studies were conducted in the United States of America (*n* = 8), China (*n* = 4), Italy (*n* = 2), Netherlands (*n* = 2), Finland (*n* = 2), and one each in Austria, France, Canada, Spain, and Australia. In four studies, only preterm infants were included, and the preterm classification ranged from 28 to 34 weeks of pregnancy. In four of the analyzed studies, gestational age was not reported, while another four reported that only full-term infants were included, although gestational age was not described. The remaining ten studies included only full-term infants born between 37 and 40 weeks of gestation. One of the studies misclassified full-term pregnancy as >35 weeks and another one included both preterm and term infants (>32 weeks). The sample size of the 23 studies ranged from 27 to 577, of which 14 studies had sample size < 100 and ten > 100. Appendix A reports the findings of studies in preterm infants and Appendix A in full-term infants.

Sequencing methodologies were surveying the 16S rRNA gene (*n* = 21), and whole-metagenome sequencing (*n* = 2) [4,19]. A single study used both sequencing methodologies (16S rRNA gene and whole-metagenome sequencing) [4]. The effect measures were mostly β and correlation coefficients. Data on neurodevelopmental outcomes were extracted from established validated scales: Ages and Stages Questionnaire (ASQ-3) (*n* = 4), the Infant Behavior Questionnaire Revised short form (IBQ-R) (*n* = 3), the Child Behavior Checklist (CBCL) (*n* = 3), the Bayley Scales of Infant and Development (BSID) (*n* = 3), the Mullen Scales of early learning (*n* = 2), the Griffiths Mental Developmental Scales (GMDS) (*n* = 2), the Behavior Assessment System for Children second edition (BASC-2) (*n* = 1), the Gesell Development Inventory (GDI) (*n* = 1), the Strange Situation (*n* = 1), and the Behavior Rating Inventory of Executive Function (BRIEF) (*n* = 1). For neural imaging, studies reported different types of magnetic resonance imaging (MRI) (*n* = 4). For functional brain network connectivity, studies used the resting state functional near infrared spectroscopy (rs-fNIRS) (*n* = 1), and for eye tracking of emotional attention, a face-distractor paradigm was applied (*n* = 1). Other studies also evaluated head circumference growth (HCG) (*n* = 1), cerebral oxygenation (*n* = 1).

### 3.2. Key Taxa Related to Neurodevelopmental Outcomes

A summary of the main associations, represented by beta coefficients and 95% confidence interval, is shown in Figure 2. This figure illustrates that among the key taxa affecting the brain, *Bacteroides* abundance during the first year of life was associated with better neurodevelopmental outcomes at 2 years of age (cognitive, language and motor scores). Similarly, *Bifidobacterium* was positively associated with emergence/extroversion (positive affectivity), higher practical reasoning, adaptive skills, and normal development scores. Varying results between Lachnospiracea, *Streptococcus*, and *Faecalibacterium* and behavior and brain development were reported.

#### 3.2.1. Bacteroides

Overall, the genus *Bacteroides* was associated with non-social fear behavior [20] and duration of orientation (orienting/regulation scale) [21]. *Bacteroides* was also associated with better cognitive, receptive/expressive languages, and motor development outcomes at 1 year of age [22,23,24]. Analyzing the effect of *Bacteroides* on brain function, this genus was positively associated with high 1 month Amygdala volume [20], fronto-parietal network connectivity at day 25 [19], being less abundant in preterm infants with suboptimal head circumference growth trajectories (SHCGT) [25].

Furthermore, we also found a cross-sectional study showing the influence of microbiome composition on behavior temperament in 3-year-old infants [26]. This study used the CBCL to evaluate children’s behavior, and *B. coprocola*, *B. nordii*, *B. coagulans* and *B. fragilis* were negatively associated with most CBCL behavior syndromes while *B. uniformis*, *B. intestinalis*, *B. caccae*, and *B. stercoris* were positively correlated with CBCL [26]. Other *Bacteroides* species (*Bacteroides vulgatus* and *B. caccae*) at 6 weeks of age were associated with better adaptive skills and hyperactivity, respectively, at 3 years of age (FDR < 0.1) [4].

#### 3.2.2. Bifidobacterium

Our systematic review found concordance between *Bifidobacterium* abundance and neurodevelopmental outcomes, such as positive emotionality [21,27], regulation, and duration of orienting [21], and social and adaptive skills [4]; in addition to an inverse association with negative emotionality [19,28]. Better outcomes for cognitive development [29,30], fine motricity [31], and protection for developmental delay in a RCT [32] were also reported. The only adverse relationship was found by Seki et al. (2021) in preterm infants with severe brain injury, although *Klebsiella* overgrowth was the main predictor for brain damage and pro-inflammatory condition [33].

#### 3.2.3. Lachnospiraceae

Findings regarding the Lachnospiraceae family were divergent across the studies, probably due to the different genera described by them. However, as the methodologies for taxonomy alignment do not always identify at the genus level within this family, we decided to keep a topic at the family level and combine these findings.

The abundance of Lachnospiraceae at 1–3 weeks of age was directly associated with surgency/extraversion at 12 months of age [27]. Additionally, *Lachnospiraceae incertae sedis* was inversely correlated with mental developmental index scores at 3 years of age [34]. In line with this finding, the Lachnospiraceae genera was highly prevalent in cases of infants with increased behavioral problems compared to the control [35]. A case-control study described Lachnospiraceae (such as *Lachnoclostridium*, *Fusicatenibacter*, *Lachnospira*, *Agathobacter* and, *Blautia*) in asphyxiated neonates correlated with ASQ-3 communication score at 6 months of [36].

On the other hand, *Tyzzerella nexilis* was associated with better depression scores among males at 6 weeks of age. At 2 years of age, the relative abundance of 4 *Blautia* ASVs were associated with worse performance in the hyperactivity scale, and these associations were more pronounced among females [4].

Increased *Robinsoniella peoriensis* and *Bacteroides* in 25-day-old infants were directly associated with fronto-parietal brain network connectivity and prefrontal cortex responses to fearful faces [19]. Finally, preterm infants with suboptimal HCG trajectories showed a decreased abundance of Bacteroidota and Lachnospiraceae from 31 to 36 weeks of age, no genera within the Lachnospiraceae family accounted for the observed change [26].

#### 3.2.4. Streptococcus

Our systematic review found the *Streptococcus* genus associated with better neurodevelopmental outcomes, such as better adaptive skills and lower incidence of internalizing problems and depression at 3 years of age [4], in addition to fewer behaviors linked to negative emotionality in 25-day-old infants [19]. In line with these findings, its abundance was found to be elevated in infants without severe brain injury [32]. The adverse findings were from the co-abundance analysis associating this taxon with mental and psychomotor developmental indexes [34] and brain volumes [20].

#### 3.2.5. Faecalibacterium

Rothenberg et al. (2021) found *Faecalibacterium* to be positively associated with better mental and psychomotor developmental indexes at 36 months of age [34]. In addition, lower prevalence of *Faecalibacterium prausnitzii* was found in infants with suboptimal HCG trajectories [25]. Conversely, *Faecalibacterium* was a dominant cluster in 1-year-old infants associated with the lowest performance in receptive and expressive language in the Mullen scale at 2 years of age [22].

#### 3.2.6. Clostridium

The *Clostridium* genus was associated with worse cognitive development and behavior. A higher abundance of *Clostridium_sensu_stricto_1* (at 1, 3, and 5 days of age) was inversely correlated with the ASQ-3 communication score [36] and several species of *Clostridium* were associated with left fronto-parietal and default mode network connectivity [19]. In addition, *Clostridium* showed a positive association with fearful faces [28] and worse adaptive skills composite scores among 3-year-old males (*Clostridium* sp. SV41) [4].

### 3.3. Alpha and Beta Diversity and Neurodevelopmental Outcomes

The Shannon index was inversely associated with negative emotionality at two months, fear reactivity at 6 months of age (domains related to an increased risk of developing anxiety disorder) [21] and inversely correlated with infant cognitive development at 2 years [22]. A higher Shannon index at 6 weeks was related to better scores for internalizing problems among males [4]. However, no associations between several metrics of alpha diversity and behavior were reported in previous studies [28,35,37,38].

Alpha diversity was positively associated with the left fronto-parietal network at 25 days, [19]. In the first year of life, reduced microbial phylogenetic diversity was correlated with increased connectivity in the left amygdala, and Chao1 had the main effect for functional connectivity between the cortex and anterior insula. A positive correlation was found only between the observed species and the connectivity of the supplementary motor area with the left parietal cortex [39].

Beta diversity at 1–3 weeks of age was related to surgency/extraversion at 1 year [27]. At 1 year of age, it showed an inverse association with non-social fear behaviors [20] and at 18 months, exhibited significant differences on fine motor activity, explaining the 4% variation in the microbiota [31]. In addition, three studies reported the influence of beta diversity on brain connectivity [20,25,29]. Beta diversity at 1 month was inversely related with prefrontal cortex volume at 1 year of age, and 1 year beta diversity was inversely related with 1 year amygdala volume [20]. Additionally, Oliphant et al. (2021) showed that beta diversity was significantly distinct between preterm infants with appropriate versus suboptimal HCG [25]. Another study in preterm infants found distinct beta diversity temporal trajectories in infants with impairment compared to infants with typical neurodevelopment [29].

### 3.4. Mechanistic Insights

While very limited functional evidence is described in humans regarding the gut microbiome influence on brain development during the first one thousand days of life, the present systematic review was able to identify findings of microbial functional pathways [4], virulence factors [19], and inflammation [32] associated with brain outcomes. Increased microbial genes encoding for virulence factors was related to increased homologous-interhemispheric connectivity [19]. Laue et al. (2021) reported several functional pathways at 6 weeks and 1 year of life associated with depression scores. Proline biosynthesis was associated with better externalizing problems, hyperactivity, and behavioral symptoms scores at 6 weeks. Catechol degradation was linked to worse performance in attention scores—catechol is a natural compound found in fruits and vegetables, but its synthetic form is frequently used as a pesticide. At 6 weeks or 1 year of age, 3 functional pathways linked to vitamin B6 were associated with better depression scores among males. The authors discuss the need of vitamin B6 for the synthesis of several neurotransmitters [4]. Both studies performed shotgun metagenomic sequencing.

Furthermore, the overgrowth of *Klebsiella* was highly predictive of brain damage and was associated with a pro-inflammatory immunological status. *Klebsiella* may drive T cell migration during the quiescent phase of brain maturation and an impaired production of neuroprotective agents accompanied by an expansion of T cells during the neurophysiological maturation phase. Mechanisms of *Klebsiella* evasion from gut are also suggested by the authors. Additionally, increased levels of fecal butyrate were observed in infants with brain injury. Butyrate fluctuations may indicate altered microbial production, as well as impaired colonocyte absorption and utilization [32]. Finally, we propose a link between mechanistic studies (results of epidemiological findings and in vitro and in vivo models) and the key taxa affecting neurodevelopment (Figure 3).

## 4. Discussion

This systematic review evaluated 23 studies reporting relationships between the early life gut microbiome and neurodevelopmental outcomes. Figure 2 illustrates that among the key taxa affecting the brain, *Bacteroides* and *Bifidobacterium* abundance were associated with better neurodevelopmental outcomes. While Lachnospiracea, *Streptococcus*, and *Faecalibacterium* presented a divergent influence on behavior and brain development.

The genus *Bacteroides* is widely present in infant gut microbiota, being an important taxon in the first year of life [40]. Several species of *Bacteroides* can metabolize human milk oligosaccharides (HMOs), mucin, and complex polysaccharides [41,42]. Additionally, functional genes related to the biosynthesis of SCFA, vitamin B6, folate, biotin and lipoic acid have been found in *Bacteroides*-enriched clusters [22,23]. Murine models show that *Bacteroides fragilis* has an important role on communicative and sensorimotor behaviors, potentially through the modulation of serum metabolites, including sphingolipids [43].

Among a variety of functions attributed to the *Bifidobacterium* genus, there is the production of lactate and acetate—end products of fermentation, which are important sources of energy for colonocytes. In gnotobiotic mice, the acetate produced by *Bifidobacterium dentium* can stimulate enterochromaffin cells to secrete serotonin (5-hydroxytryptamine [5-HT]) [44], an important stimulator of the enteric nervous system and GI function [45]. *Bifidobacterium* also produces other neuroactive substances, such as y-aminobutyric acid (GABA), an inhibitory neurotransmitter that can act directly on the enteric nervous system [46].

Neonatal gut colonization by *Bifidobacterium* is responsible for immune maturation in childhood. The biosynthesis of acetate by Bifidobacteria, is known to drive Treg cell differentiation and protect against infection by improving gut barrier integrity [47]. Immunomodulatory bacterial products can access the brain and influence neurophysiology, particularly microglia and astrocytes. In early life, microglia conduct synaptic transmission due to their capacity for synaptic pruning through the neuronal circuit, while astrocytes are essential for the maintenance of the BBB and modulation of brain metabolism [4].

While *Bacteroides* and *Bifidobacterium* were more frequently associated with better developmental outcomes, findings related to Lachnospiracea, *Streptococcus*, and *Faecalibacterium* demonstrated a variable influence on behavior and brain development. The Lachnospiraceae family includes 58 genera of phylogenetically and morphologically heterogeneous taxa, such as *Blautia*, *Coprococcus*, *Dorea*, *Lachnospira*, *Oribacterium*, *Roseburia*, and *Ruminococcus*, among others. The influence of this family on human health is generally related to the production of SCFAs. SCFAs are important in preserving the integrity of the intestinal barrier through the regulation of tight junction proteins and similar effects of SCFA are described in the BBB [48]. *Streptococcus* forms an important trophic chain with other gut bacteria, including *Bifidobacterium*, by producing lactate and propionate and is related to the production of serotonin [46].

Modulation of the gut–brain axis by the microbiome metabolites (SCFA, as butyrate, propionate, and acetate) is linked to function and maturation of specific cell types in the CNS, including microglial cells [49,50]. In addition, butyrate is involved in neurotransmitter release mechanisms [51], as a facilitator of synaptic plasticity and processes related to cognition, learning and memory [52], and propionate can influence gap junction gating in the BBB [53]. Therefore, it has been suggested that SCFAs facilitate the proliferation of neural precursor cells, a possible mechanistic insight of how maternal gut microbiota may influence embryonic brain development [54]. In humans, butyrate prevented maternal diet-induced neurocognitive deficits in newborns [52].

Microbiota products may also be one of the players in recent neuroimmune hypotheses of neuropsychiatric disorders. It is known that some metabolites may trigger the host’s immune response leading to the production of cytokines and other inflammatory mediators that can reach the brain through mechanisms involving changes in BBB permeability [55]. In vitro, *Faecalibacterium prausnitzii* has been associated with the increased production of IL-10 and IL-12 [56,57]. Additionally, an in vivo model using gnotobiotic mice found that salicylic acid by *F. prausnitzii* showed anti-inflammatory effects by blocking the production of IL-8 [57]. In this scenario, microbiota products or their effects in the periphery (e.g., via immune system) could directly reach the brain via afferent neurons in the vagal nerve, through the release of neurotransmitters in the gut. In germ-free mice, the vagus nerve appears to be critical for *Lactobacillus rhamnosus*-reduced stress-induced anxiety and depressive-like behaviors and GABA-A receptors expression in the brain [58].

Our review noted that increased abundance of *Bifidobacterium* and *Klebisiella* were found in infants with brain injury [32]. Moreover, *Faecalibacterium* was related to worse language performance on cluster analysis [22]. Increased abundance of beneficial bacteria in cases of neuroimpairment has been previously reported in toddlers [59], adults and elderly patients [60]. Interestingly, increases in the abundance of beneficial bacteria have been observed in other studies describing different pathological conditions, such as gestational diabetes mellitus [61]. We can hypothesize that the abundance of beneficial bacteria increases in pathological conditions to mitigate microbiome dysbiosis.

Although the key players in the microbiome have been well described across the studies analyzed herein, the functional pathways involved remain to be elucidated. Laue et al. (2021) shed some light reporting NAD salvage (pathway for the synthesis of nicotinamide adenine dinucleotide), aspartate and asparagine biosynthesis, methanogenesis, bile acid transformation pathways, among others, to be related to behavioral outcomes [4]. We hypothesize that the decreased bile acid transformation pathway, associated with depression and the adaptive skills score [4] leads to lower resistance to commensal species colonization and pathogenesis, as previously described in gnotobiotic mouse model [62]. Another important pathway affected was the production of asparagine and aspartate via glutamate degradation, as the depletion/increase of an excitatory neurotransmitter can affect behavior.

Study limitations include different classifications of preterm delivery or missing information. Preterm delivery shapes the intestinal environment [63] and is associated with lower scores in cognition tests [31]. Thus, appropriate collection and modeling of information on this variable is recommended for infant microbiome studies. We also found adjustments for confounding variables to be different among the studies. While several studies included many variables in the adjusted models, others did not adjust for potential confounders [26,32] or multivariate analysis was not described [29]. These factors make comparisons between studies difficult and limit the conclusion of this systematic review.

The variation in the results across studies may also reflect the dynamic composition of the infant gut microbiome [64]. Thus, the age of microbiome sampling can be considered in the analysis and when comparisons between the studies are being performed. This variation may also reflect different susceptibility windows in neurodevelopment [4]. Our systematic review found that the age of microbiome evaluation was important for significant associations, except in the study by Ou et al. (2022) who found no significant association in early microbiome assessment (1, 3, and 4 months), but only at childhood (6 years) [38]. Similarly, Eckermann et al. (2022) analyzed the microbiome at the same time points and no associations were found in infancy and childhood [37].

Although not consistently, sex differences were observed in many studies. Sex differences in the associations between *Bacteroidetes* and cognitive and language development [23], surgency, regulation, duration of orienting, cuddliness, and fear reactivity [21]. Several other altered taxa including *Clostridium*, *Parabacteroides*, Lachnospiraceae, *Collinsella*, and *Citrobacter* were associated with face bias [28]. Laue et al. (2021) reported sex-specific effects in nearly all associations, suggesting that males might be more susceptible [4]. By contrast, Zhang et al. (2021) found no changes when adjusting for sex [36].

These differences across studies could arise from differences during neurodevelopment attributed to sex, or spuriously due to small sample sizes of sex-specific analyses [4]. Supporting the first suggestion, Rothenberg et al. (2021) described males with lower psychomotor scores compared to females [34]. Sex was also associated with cognitive and language scores [23]. Limited sample size was an issue in most of the studies (15 studies had a sample size < 100). A large enough sample size is needed for statistical power and to show an effect of a given magnitude [65]. Simultaneous adjustment for the covariates discussed above also requires an adequate sample size.

In the present systematic review, we discuss the key taxa and their functions when interacting with the enteric nervous system, immune system, autonomic nervous system, and how it may impact the developing brain (Figure 3), suggesting the microbial mechanisms that influence brain maturation and can serve as a target for intervention during the first 1000 days of life. Furthermore, understanding the essential role of early gut microbial colonization, its produced host interactions and metabolites can lead to a better comprehension of the microbiome–gut–brain axis.

Further studies evaluating the functional capacity of the microbiome would improve our understanding of the roles of early gut microbiome in healthy neurodevelopment. In addition, advances in data integration analyses, method harmonization, and multicentric studies using integrative omics approaches will change the current state-of-the-art in how and why specific taxa, metabolic pathways, and metabolites produced by gut bacteria influence the process of synaptic neuroplasticity, neural connectivity, neurogenesis, and the maturation of the human brain.

## 5. Conclusions

There is sufficient evidence that the gut microbiome is a relevant factor for the course of neurodevelopment, including behavioral and cognitive function outcomes. *Bacteroides* and *Bifidobacterium* were more frequently associated with non-social fear behavior, duration of orientation, cognitive and motricity development, and brain development. While results related to Lachnospiracea, *Streptococcus*, and *Faecalibacterium* demonstrated a variable influence on behavior and brain development. The microbial mechanisms reported were related to NAD, aspartate, and asparagine synthesis, methanogenesis, bile acid transformation pathways, microbial virulence factors, neurotransmitters, short chain fatty acids production, and immune mediators.

## Figures and Tables

**Figure 1 microorganisms-12-00424-f001:**
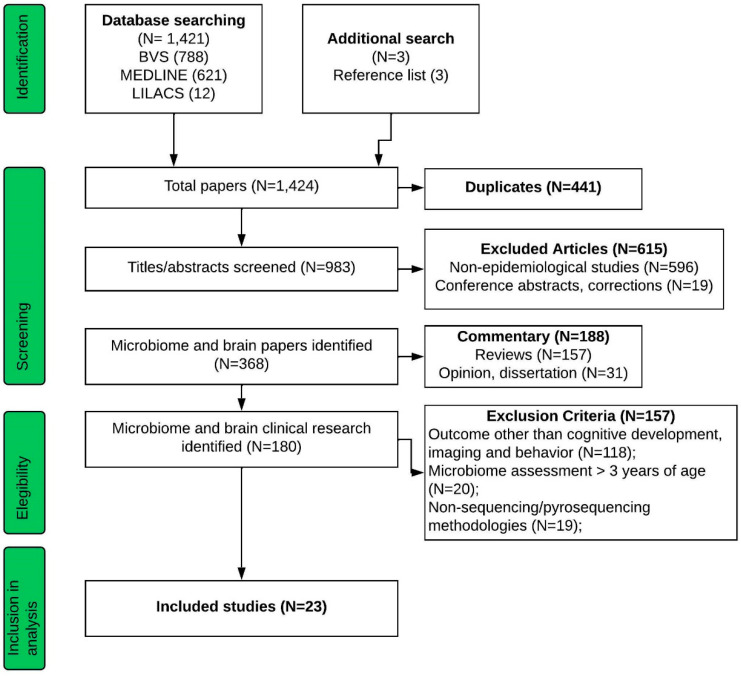
Flow Diagram of Database Search and Eligibility (modified from PRISMA flow diagram).

**Figure 2 microorganisms-12-00424-f002:**
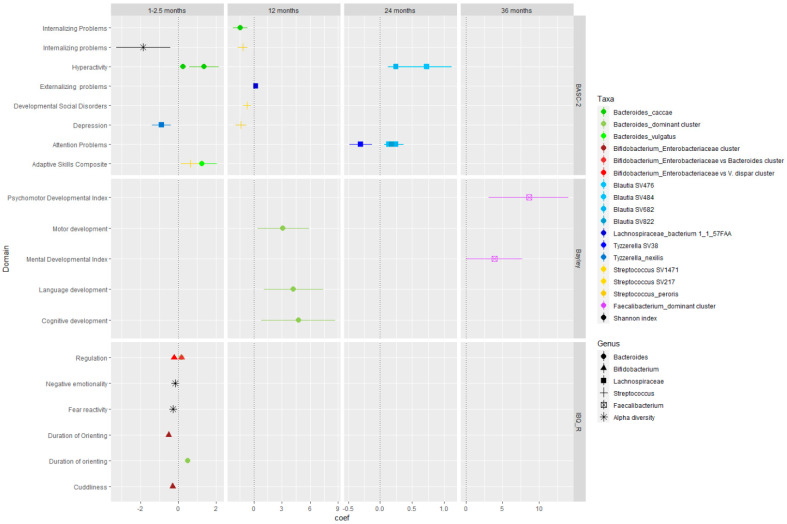
Early life gut microbiome associations with neurodevelopmental outcomes across time. Associations reported by multiple studies between abundances of *Bacteroides*, *Bifidobacterium*, Lachnospiraceae, *Streptococcus*, *Bifidobacterium*, and *Faecalibacterium* and neurodevelopmental outcomes. The point indicates the β estimate reported by studies and the horizontal band represents the 95% confidence interval; Colors are bacterial species, clusters description and alpha diversity (legend on the top right side); Symbols are genus and alpha diversity denominations (legend on the bottom right side). BASC-2: Behavior Assessment System for Children second edition; IBQ-R: Infant Behavior Questionnaire Revised short form.

**Figure 3 microorganisms-12-00424-f003:**
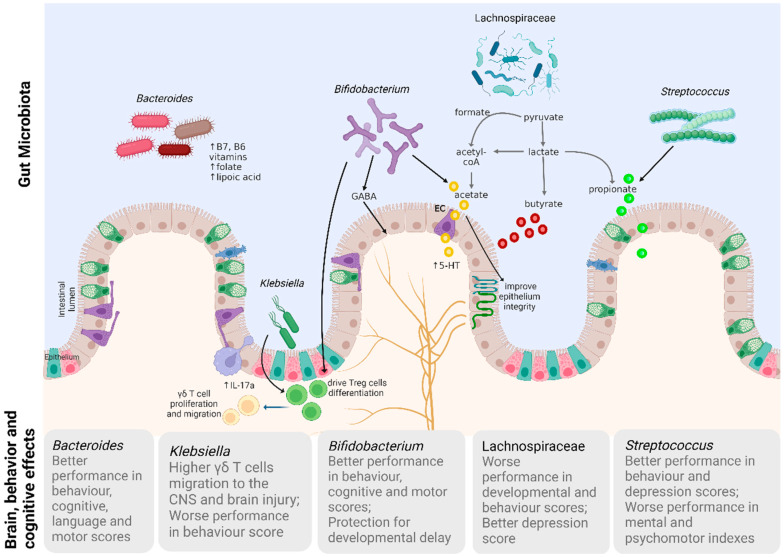
The role of gut microbiota composition on brain development during the first one thousand days of life. The gut microbiota can impact the host enteric and central nervous system through various mechanisms that collectively may affect the brain, behavior, and cognition. These mechanisms include the biosynthesis of vitamins, folate, and lipoic acid related to the abundance of *Bacteroides*. *Bifidobacterium* plays a role by producing acetate and neuroactive substances, such as GABA and 5-HT. Also, *Bifidobacterium* can induce immune cells maturation and improve the epithelium integrity (effect described in the BBB as well). *Bifidobacterium*, Lachnospiraceae, and *Streptococcus* integrate a trophic chain to produce the main SCFAs, acetate, butyrate, and propionate. *Klebsiella* is a potential pathogen genus, its abundance is linked to a pro-inflammatory T cell response and brain injury. SCFA: short chain fatty acid; BBB: blood–brain barrier; 5-HT: 5-hydroxytryptamine; GABA: gamma aminobutyric acid. Upward arrows indicate increase of B7, B6, folate, lipoic acid, IL-17a and 5-HT. Figure created with BioRender.com.

## Data Availability

Data are contained within the article and Appendix A.

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
