# Peer review of "The Gut Microbiome in the First One Thousand Days of Neurodevelopment: A Systematic Review from the Microbiome Perspective"

_microorganisms, 2024, doi:10.3390/microorganisms12030424_

Round 1

Reviewer 1 Report (Previous Reviewer 1)

Comments and Suggestions for Authors

Reviewer 2 Report (New Reviewer)

Comments and Suggestions for Authors

The manuscript titled: The gut microbiome in the first one thousand days of neurodevelopment: A systematic review from the microbiome perspective
is an interesting review that targets the microbiota populations in link with brain health and disorders. The authors did great efforts in writing this review in a well-structured way and presenting important data. In addition, highlighting the association between some microbiome strains such as Bacteroides and Bifidobacterium with non-social fear behavior as an example of microbiome-gut-brain axis.

I recommend the acceptance of the review titled: The gut microbiome in the first one thousand days of neurodevelopment: A systematic review from the microbiome perspective.  The evaluation is based on that overall merit of the review scope and highlights that have addressed. 

Reviewer 3 Report (New Reviewer)

Comments and Suggestions for Authors

The authors of the manuscript describe the results of a systematic review of 23 studies that reported on the relationship between the gut microbiome at an early age and the results of the development of the nervous system.

The systematic review was carried out carefully and at a good methodological level. Despite the limitations of the systematic review mentioned by the authors of the manuscript, it has important scientific significance. The results shown by the authors of the manuscript may be of interest to researchers in the field of pediatrics, neurobiology and neurology. Therefore, the conducted research will be relevant for publication in the journal Microorganisms.

This manuscript is a resubmission of an earlier submission. The following is a list of the peer review reports and author responses from that submission.

Round 1

Reviewer 1 Report

Comments and Suggestions for Authors

short the introduction, and make a concise and deep introduction, rather than a speak generally introduction.

The scientific database was consulted up to May 2022.  I suggested authors inclused all articles till now, more than 1 years span, so this will lost many important papers in this field

The paper is more like a meta anysis, rather than a review. I suggest authors choose a suitable type

Similar work has been published, and this paper can not provide authors new understanding to readers. Moreover, authors only focus on the changes of bacteria, lack deep mechanisms and their own understanding on this field.

Comments on the Quality of English Language

short the introduction, and make a concise and deep introduction, rather than a speak generally introduction.

The scientific database was consulted up to May 2022.  I suggested authors inclused all articles till now, more than 1 years span, so this will lost many important papers in this field

The paper is more like a meta anysis, rather than a review. I suggest authors choose a suitable type

Similar work has been published, and this paper can not provide authors new understanding to readers. Moreover, authors only focus on the changes of bacteria, lack deep mechanisms and their own understanding on this field.

Reviewer 2 Report

Comments and Suggestions for Authors

Even though the idea sounds interesting, there are some important points that need clarification, refinement, reanalysis, rewriting, and more detailed information to improve this article.

Major points

1.       The manuscript needs writing editing. It would be better to improve the title, it should reflect the content of this experimental study, for example "…. A Systematic Review”. The main objective must be the same throughout the manuscript (abstract, introduction and results/discussion), for example, “Here, we performed a systematic review of gut microbiome assessments in infants (healthy and with brain pathology) during the first thousand days of life, associated with neurodevelopmental outcomes. Also, we investigated the key actors and mechanisms of the microbiome that affect this development. “Authors should not use the words that appear in the title as keywords. References must be recent and relevant.

2.       The introduction section: The introduction section of this systematic review is too long and should be improved and rewritten. Why is this study review important? A proper presentation and a good and clear justification (reason) for conducting this review study should be given. It would be better if this section had no more than 5 paragraphs covering the topics to be covered (neurological development in the first 1000 days, most prominent gut microbiome, neurological problems due to external factors including the microbiome). The research question should be clearly outlined. It would be better if the authors offered the hypothesis before the main objective of this study. What is the main aim of this study? The main aim/objective/goal of this study must be the same throughout the manuscript.  

3.       The materials and methods section needs improvement. It would be better if the authors specified more about the inclusion and exclusion criteria that were used. More details are needed to characterize this criterion. What kind of studies were included? Why did not the authors use other review studies? The authors should describe which validated scales/tools would be accepted to assess children's neurodevelopment. Did the authors consider using any statistical analysis? Who conducted this study? What ethics committee approved this study? When reporting on research that involves human subjects, human material, human tissues, or human data, authors must declare that the investigations were carried out following the rules of the Declaration of Helsinki. Did the authors consider using any statistical analysis? Authors should report what kind of comparisons were made. How did the authors evaluate the gut microbiota associated with neurodevelopment during the first 1000 days? All parameters studied should be described, defined, and measured appropriately. This section must provide enough detail on the study design for it to be replicable.

4.       In the results section: all parameters that the authors searched for and collected should be described in detail in the material and methods section. It is not necessary to add a theoretical introduction to the results in each subsection. It would be better to use them in the introduction/discussion section. The text should provide a better summary of the most significant results and avoid repeating the same information in the text if this data appears in the tables. In tables, authors should use a legend for abbreviations.

5.       The discussion needs deep improvements. This section should start with the main objective of this study and the most significant results. The discussion should be more argumentative about the main objective and the most significant results of this study. The results must be discussed from multiple angles and placed in context without being over-interpreted. It should be clear why this study is important and how it improves existing knowledge on the subjects covered. A paragraph of strengths/limitations and suggestions for this study should be written before the conclusion.

6.       The conclusion section: The conclusion should be the same throughout the manuscript.

 The manuscript is difficult to read. The main objective is not clear. There are not enough details about the study design of this systematic review to be able to replicate it. The intro section is too long. The results section does not reflect what results the authors used to answer the main objective. Apart from tables 1 and 2, it would be good if the authors showed the results comparatively by neurodevelopmental parameter. Authors should avoid writing what is already seen in Tables 1 and 2. They should guide readers toward the goal of this review. That is, associating the intestinal microbiota with neurological development. So, in both the results and discussion sections, they should show them in subsections according to the specific assessment, for example, non-social fear behavior, orientation duration, cognitive and motor development, brain development, etc. I encourage the authors to rewrite the manuscript, thinking about the principal goal of this systematic review, and its design and answering with the results and arguments of the discussion the most proper conclusion to this research work.

Comments on the Quality of English Language

 Minor editing of English language required.